# Neurons and Glia Interplay in α-Synucleinopathies

**DOI:** 10.3390/ijms22094994

**Published:** 2021-05-08

**Authors:** Panagiota Mavroeidi, Maria Xilouri

**Affiliations:** Center of Clinical Research, Experimental Surgery and Translational Research, Biomedical Research Foundation of the Academy of Athens, 11527 Athens, Greece; pmavroeidi@bioacademy.gr

**Keywords:** aggregation, astrocytes, a-Synuclein, inclusions, microglia, neurons, oligodendroglia, seeding

## Abstract

Accumulation of the neuronal presynaptic protein alpha-synuclein within proteinaceous inclusions represents the key histophathological hallmark of a spectrum of neurodegenerative disorders, referred to by the umbrella term a-synucleinopathies. Even though alpha-synuclein is expressed predominantly in neurons, pathological aggregates of the protein are also found in the glial cells of the brain. In Parkinson’s disease and dementia with Lewy bodies, alpha-synuclein accumulates mainly in neurons forming the Lewy bodies and Lewy neurites, whereas in multiple system atrophy, the protein aggregates mostly in the glial cytoplasmic inclusions within oligodendrocytes. In addition, astrogliosis and microgliosis are found in the synucleinopathy brains, whereas both astrocytes and microglia internalize alpha-synuclein and contribute to the spread of pathology. The mechanisms underlying the pathological accumulation of alpha-synuclein in glial cells that under physiological conditions express low to non-detectable levels of the protein are an area of intense research. Undoubtedly, the presence of aggregated alpha-synuclein can disrupt glial function in general and can contribute to neurodegeneration through numerous pathways. Herein, we summarize the current knowledge on the role of alpha-synuclein in both neurons and glia, highlighting the contribution of the neuron-glia connectome in the disease initiation and progression, which may represent potential therapeutic target for a-synucleinopathies.

## 1. Introduction

The presynaptic neuronal protein alpha-synuclein (aSyn) under physiological conditions regulates neurotransmitter release and SNARE (soluble N-ethylmaleimide-sensitive factor attachment protein receptor) complex assembly and is considered a chameleon-protein due to its remarkable conformational plasticity [1]. On the other hand, aggregated aSyn is the major component of the proteinaceous inclusions found in the degenerating neurons of Parkinson’s disease (PD) and dementia with Lewy bodies (DLB) brains, known as Lewy bodies (LBs) and Lewy neurites (LNs) [2]. Alpha-synuclein also has a strong genetic link to PD pathogenesis, since missense point mutations of the *SNCA* gene encoding for aSyn, *SNCA* gene locus duplications and triplications or gene-enhanced expression are the main causes of familial PD [3,4,5,6,7,8,9]. In contrast, multiple system atrophy (MSA), a fatal debilitating neurodegenerative disorder, is characterized by the presence of aggregated aSyn within the glial cytoplasmic inclusions (GCIs) present in the cytoplasm of oligodendrocytes [10,11,12]. Glial aSyn accumulation is also evident in PD and PD with aSyn-positive deposits reported in astrocytes and oligodendrocytes [13,14,15]. Contrarily, aSyn-positive inclusions in astrocytes have been also found in MSA [16], but to a lesser extent [14] compared to neuronal and oligodendroglial inclusion pathology.

The clinical and neuropathological heterogeneity in a-synucleinopathies may ascend from the unique properties of the different conformational aSyn strains found in neurons or glia that might contribute to distinct clinical phenotypes [17,18,19,20]. Even though the physiological and pathological functions of aSyn in neurons, where the protein is physiologically expressed, are well characterized, the mechanisms underlying the pathological accumulation of aSyn in the glial cells of the central nervous system (CNS) still necessitates further investigation. Microglia and astroglia have distinct roles in maintaining brain’s homeostasis but under stress conditions, such as increased aSyn burden, they can become activated and contribute to disease pathology by triggering neuroinflammatory mechanisms. Reactive astrocytes and microglia have been detected in human post-mortem brains of a-synucleinopathies [13,21,22,23,24,25], further supporting a role of active gliosis in the initiation and progression of the disease. Moreover, all glial cells have been reported to internalize aSyn and a neuron-to-glia transmission is thought to underlie the propagation of aSyn pathology in a-synucleinopathies. In the following sections, we discuss how aSyn affects neuronal and glial function and homeostasis in health and disease.

## 2. Alpha-Synuclein in Neurons: A Multifaceted Protein

### 2.1. A Role at the Synapse

alpha-Synuclein (aSyn) is a small, intrinsically disordered protein that is mainly localized at the pre-synaptic terminal [26,27], but is also present in the neuronal somato-dendritic compartment [28], in red blood cells [29], in the gut and other peripheral tissues [30,31,32]. Although aSyn is highly enriched in presynaptic boutons, it displays a delayed distribution in the terminals, suggesting that it is implicated in later stages of synaptic development, rather than playing a central role in synapse modulation [27]. Importantly, aSyn is differentially expressed in the various neuronal cell types, being more abundant in excitatory synapses across different brain regions and particularly in central catecholaminergic systems [33]. On the contrary, the protein displays a differential expression profile in inhibitory synapses amongst the different brain areas, with a particular interest of aSyn presence in striatal GABAergic medium spiny neurons (MSNs) [34,35].

The first indication regarding the role of aSyn on neural plasticity arose about 25 years ago, when “synelfin” (synuclein, NACP) expression was found up-regulated during bird song learning [36]. The localization of aSyn in pre-synaptic boutons is mainly attributed to its tight association with synaptic vesicle membranes [37] and its high affinity for the SNARE complex proteins synaptobrevin-2 (or Vesicle Associated Membrane Protein 2, VAMP2), synapsin III and rab3A [38,39,40]. It has been proposed that aSyn interacts with VAMP2 and promotes SNARE complex assembly [38], followed then by its disassembly in order to complete the round of membrane fusion (Figure 1). The crucial role of aSyn assembly with SNARE complex on neuronal survival was further verified by the neuronal dysfunction and impaired survival of triple αβγ-synuclein knockout mice during ageing [38,41]. Interestingly, aSyn lentiviral overexpression in primary neurons led to enhanced SNARE complex assembly, further supporting the role of this protein in synaptic activity [38]. The same group later showed that only multimeric membrane-bound, but not the soluble monomeric aSyn, can promote the SNARE complex assembly [42]. It has been also recently suggested that aSyn is involved in synaptic vesicle homeostasis at the pre-synaptic terminal via a calcium (Ca^2+^)-dependent mechanism [43].

On the contrary, natively unfolded monomeric aSyn at the pre-synaptic terminal is prone to form pathological conformations, thus exerting neurotoxic effects [44] (Figure 1). It has been additionally suggested that aSyn is preferably bound to synapsin 1 and VAMP2 when the protein is present in its oligomeric form [45], highlighting the importance of the conformational state of aSyn for its proper function. There are also findings supporting the implication of aSyn in synaptic transmission, due to its association with the synaptic vesicle pool, modulating the vesicle mobility, the recycling pool homeostasis and endocytosis [46,47,48].

Alpha-synuclein can also function as a molecular chaperone via effective binding to other intracellular proteins. The first indication came with the discovery that aSyn displays structural and functional homology with other molecular chaperones, as the 14-3-3 or small heat shock proteins [49,50]. Additional studies revealed that aSyn synergistically acts with the presynaptic cysteine-string protein-alpha (CSPalpha) promoting the assembly of the SNARE complex [38,51], further validating its chaperoning properties. Biochemical and structural analysis of aSyn strengthened the current indications for its chaperone-like function via its C-terminal region (residues 61-140) [52,53,54]. However, following studies indicated that the chaperone-binding site of aSyn lies within the non-amyloidal component (NAC) region (residues 61-95), which is prone to aggregation and thus highly susceptible to form fibrils [55,56].

### 2.2. Association with Membranes and Lipid Trafficking

Intracellular aSyn can be found either natively unfolded in a soluble state or membrane-bound forming an alpha-helical or a beta-sheet secondary structure, depending on the solution conditions [57,58,59]. It has been proposed that there is a bidirectional link between aSyn species formation and membrane remodeling, meaning that not only aSyn structure is affected upon lipid interaction, but also that membrane integrity depends on the presence of different aSyn conformations [60,61,62]. However, there are controversial results regarding the association of aSyn with membrane lipids and its conformational state, with some studies reporting that membrane-bound aSyn gets protected from aggregation, thus leading to neurotoxicity attenuation [44,63,64], whereas others suggest that interaction of aSyn with membranes triggers its self-association and subsequent aggregation [65,66,67]. Importantly, it has been shown that the PD-related aSyn mutations reduce its interaction with membranes, thus further suggesting that aSyn binding on membranes may exert neuroprotective effects [68,69,70,71,72].

A plethora of studies argue that aSyn in its soluble state exists as a monomer [73,74,75,76], whereas others suggest that it occurs physiologically as a tetramer resisting aggregation [77,78,79]. In the presence of lipid membranes, aSyn adopts an alpha helical structure in the N-terminus region that stabilizes the formation of high-order aSyn multimers [42,73,80,81]. Interestingly, the membrane curvature seems to affect the structure of aSyn, which can adopt either an elongated or a broken alpha-helix conformation, when bound to a large diameter (∼100 nm) or a small, highly curved vesicle, respectively [82,83,84,85]. It has been also proposed that aSyn has a role in lipid metabolism, since it participates in fatty acids transportation between the cytosol and membranous compartments [86,87] and in lipid and membrane biogenesis organizing and stabilizing the lipid bilayer of membranes and vesicles [88,89]. On the other hand, disrupted aSyn expression pattern leads to lipid dysregulation, since both the absence and the overexpression of either wild-type (WT) or mutated aSyn gives rise to abnormal lipid metabolism [90,91,92,93]. Finally, several studies have demonstrated that aSyn regulates membrane homeostasis via inhibition of phospholipases activity, such as phospholipase D [94,95,96,97]; however, there are controversial results in the literature [98].

### 2.3. Aggregation and Post-Translational Modifications

alpha-Synuclein is composed of three distinct domains: the N-terminal lipid-binding domain, the NAC region and the C-terminal binding domain [84,99,100]. A central role in the fibril formation and subsequent aggregation of aSyn is thought to be mediated through the NAC region of the protein composed of nonpolar side-chains and assembles cross b-structures. Based on that, it has been shown that the deletion of specific residues (74-84) within the core region can abolish aSyn aggregation [101,102]. It has been also demonstrated that the endogenous neuronal aSyn and the interaction of aSyn with lipids plays a central role for aSyn recruitment and subsequent seeding of pathology, as it could behave as a core for the formation of insoluble aggregates [35,75,103,104].

Several mutations in the *Snca* gene have been linked to PD pathogenesis, such as the A53T, A30P, E46K, H50Q, G51D, A18P, pA29S and A53E mutations, all located in the N-terminus region [3,5,7,68,105,106,107]. Most of them are tightly linked to enhanced aSyn aggregation, pathology progression and clinical manifestations in PD. Specifically, A53T and A30P aSyn mutants are natively unfolded, similarly to WT protein. However, at higher concentrations A53T has been shown to accelerate aSyn fibrillization, a critical event in PD pathogenesis [108,109,110]. On the other hand, A30P promotes aSyn oligomerization rather than fibrillization, thus reducing aggregate formation [109,111]. The E46K mutation leads to conformational changes of aSyn due to C-terminal to N-terminal contacts in the monomeric protein, resulting in enhanced aSyn accumulation [111,112,113]. Moreover, the PD-linked H50Q point mutation increases aSyn aggregation propensity and toxicity [114], whereas the G51D mutation has the opposite effects [115]. However, although G51D mutants seem to oligomerize in a slow rate, they form more toxic fibrils, thus suggesting distinct disease mechanisms for the various aSyn mutations [116,117]. Similarly, A53E mutant seems to lead to neuronal toxicity via an aSyn aggregation-independent manner [118]. Strikingly, the G51D and A53E aSyn mutations have been proposed as potential links between PD and MSA [106,119]. However, up-to-date, no hereditable mutations in the coding region of *SNCA* gene have been identified in MSA cases [120]. Apart from point mutations [117,121,122], various post-translational modifications are implicated in aSyn aggregation, the most important of which are phosphorylation, sumoylation, ubiquitination, nitration, N-acetylation, O-GlcNAcylation and truncation.

The phosphorylation of aSyn both at serine and tyrosine residues and particularly at Ser129 is widely considered as an indicator of pathology. However, the effect of Ser129 phosphorylation on aSyn toxicity is still under debate, with the majority of studies suggesting that it accelerates cell toxicity and neurodegeneration [123,124,125,126,127]. Contrarily, others have proposed a neuroprotective role of Ser129 phosphorylation since it was reported to drive the conversion of toxic oligomers into less harmful aggregates [128,129,130]. Other mechanisms of phosphorylated Ser129 aSyn-mediated neuroprotection include inhibition of its fibrillation [131], upregulation of tyrosine hydroxylase (TH) activity [132] or lowering of the protein’s membrane-binding affinity [133]. Although the 90% of aSyn in LBs is found phosphorylated at Ser129, a significant amount of phosphorylated Ser129 aSyn is also detected in a soluble, rather than in an aggregated state in PD brains [134], whereas only a small percentage of aSyn is phosphorylated at Ser129 in the brains of healthy controls [135,136,137]. In addition, aSyn can be phosphorylated at Ser87, Tyr125, Tyr133 and Tyr136 residues [138,139] and these are also implicated in either neurotoxic or neuroprotective events [127,138,140,141]. Nonetheless, in most in vivo models where aSyn is overexpressed (virally, transgenic or PFF-inoculations) the detection of pSer129 positive aSyn signal is invariably linked to neurotoxicity, indicating a rather neurotoxic and not a neuroprotective role.

Nitrated aSyn is also tightly linked to neurodegeneration, as demonstrated by experiments in both cellular and animal models, as well as in patient-derived brains [142,143,144,145], through its implication in oxidative damage and disease development [146]. Four tyrosine residues in aSyn sequence, Tyr39 (within the N-terminus), Tyr125, Tyr133 and Tyr 136 (within the C-terminus) can undergo nitration. Nitration at Tyr39 has been shown to result in low binding affinity of aSyn on lipid vesicles due to its loss-of-alpha helical conformation status [147], whereas nitration at Tyr125 seems to play a crucial role for aSyn dimerization [148]. Moreover, the linking between two tyrosines is considered as a potential mechanism for aSyn oligomer stabilization and its subsequent aggregation into proteinaceous inclusions [149]. In addition, the detection of nitrated aSyn in the human blood serum could potentially serve as a clinical biomarker for PD diagnosis [150].

Another aSyn post-translational modification crucial for its aggregation propensity is ubiquitination, via regulation of the proteasome-dependent protein degradation [151] and the subcellular localization of the protein [152]. Ubiquitinated aSyn has been isolated from LBs and sarkosyl-insoluble fractions derived from synucleinopathy brains [153,154]. CHIP (C-terminal U-box domain of co-chaperone Hsp70-interacting protein), SIAH (seven in absentia homolog) and Nedd4 (neuronal precursor cell-expressed, developmentally down-regulated gene 4) have been identified among the E3 ubiquitin ligases implicated in aSyn ubiquitination [155,156,157,158,159,160]. Ubiquitin modification has been demonstrated to have differential effects on aSyn accumulation and subsequent aggregation, dependent on the residue being modified. More precisely, ubiquitination at Lys6, Lys12 and Lys21 residues has been shown to moderately inhibit aSyn fibrillation, whereas at Lys10 and Lys23 residues has been reported to promote the formation of aSyn inclusions [161]. In addition, ubiquitination at Lys32, Lys34, Lys43 and Lys96 inhibits aSyn aggregation [161].

Sumoylation is a similar process to ubiquitination, since aSyn is conjugated to SUMO (small ubiquitin-like modifier) at lysine residues. SUMO-1 was found in aSyn-positive inclusions of a-synucleinopathy brains or associated with lysosomes of PD animal models [162,163,164]. It has been also suggested that aSyn sumoylation facilitates its aggregation since it inhibits its degradation [165], whereas other studies proposed a neuroprotective role of aSyn sumoylation, which seems to promote aSyn solubility and thus inhibit its aggregation [166,167]. The discrepancy between these data may be attributed to the different lysine residues available for sumoylation being investigated in each study. Another aSyn modification that has been up for debate is its N-terminal acetylation. Although many studies have assigned a neurotoxic role on aSyn N-acetylation, as it has been shown to promote aSyn β-sheet formation and fibrillation [168,169,170], others suggest that either N-acetylated aSyn mediates its physiological binding on synaptic vesicles [171], or it acts in a protective manner against aSyn aggregation [172,173].

O-GlcNAcylation is a biochemical process that involves the attachment of O-linked N-acetylglucosamine to Ser and Thr residues of various proteins, amongst which is aSyn. Murine and human aSyn have been shown to be O-GlcNAcyled in many threonine residues including Thr33, Thr34, Thr54, Thr59, Thr64, Thr72, Thr75, Thr81 and Thr87 [174,175,176,177,178] and this post-translational modification has repetitively been linked to reduced aSyn aggregation and attenuation of PD-related toxicity [179,180,181,182]. Finally, aSyn truncation has gained scientific attention, given that C-terminally truncated aSyn has been identified in the inclusions present in PD brains [183,184,185]. Many studies have considered that aSyn truncations have neurotoxic effects due to increased accumulation of misfolded aSyn [186,187,188,189,190,191,192,193,194].

### 2.4. Channel Formation/Channel Interactions

As mentioned above, membrane-bound aSyn adopts an alpha-helical conformation, which facilitates its oligomerization and subsequent aggregation. It has been suggested that aSyn oligomers can form transmembrane channels and pore-like structures that have been linked to pathological events during PD development (Figure 1) [195,196,197]. As a result, vesicles or low-molecular mass molecules may penetrate the cell membrane and in combination with altered cellular ionic homeostasis could potentially lead to cell toxicity and neuronal degeneration [198,199]. Another mechanism for the increased membrane permeability involves the incorporation of aSyn oligomers between the membrane phospholipids, thus leading to the bilayer thinning which thereafter allows the diffusion of small molecules [200].

A wide range of studies has demonstrated that the ion channels formed by oligomeric aSyn dysregulate cellular ion concentrations and may represent a critical event in the pathogenesis of a-synucleinopathies [198]. Some PD-linked aSyn mutations, such as E46K and A53T, have been shown to be implicated in the channel formation, whereas other aSyn mutants (i.e., A30P) have displayed low membrane affinity [197,201]. However, other groups have shown that A30P and A53T aSyn mutations are responsible for the formation of large membrane pores through which most cations can pass non-selectively [202]. It has been reported that the formation of such cation-permeable pores could lead either to ion conductivity or to increased Ca^2+^ influx and subsequent cell death [202,203,204,205]. Upon aSyn cation channel opening, other channels, such as the ATP-dependent potassium channels K (ATP), have been reported to be activated in hippocampal neurons and this could probably diminish the aSyn-dependent neuronal excitability [205].

Binding of aSyn to the plasma membrane results in the formation of aggregates and this aggregation leads to the redistribution of the α3 subunit of Na^+^/K^+^-ATPase. As a result, Na^+^/K^+^-ATPase is no longer able to effectively pump out Na^+^ from neurons, thus leading to an intracellular Na^+^ accumulation [206]. Furthermore, extracellular aSyn was reported to activate the voltage-gated Ca^2+^ channel Cav2.2 in rat neurons, due to disorganization of lipid rafts in the plasma membrane, resulting in enhanced dopamine release and increased Ca^2+^ influx [207]. Both events may explain the synaptic dysfunction and neuronal vulnerability in PD. L-type Ca^2+^ channels are also implicated in PD development, as administration of L-type Ca^2+^ channel blockers (i.e., isradipine, nimodipine) in animal models and PD patients, reduced death risk and ameliorated disease manifestations [208,209,210,211]. Finally, aSyn oligomers can inhibit α4β2 nicotinic acetylcholine receptors of dopaminergic neurons, thus leading to cholinergic signaling deficits [212]. In summary, aSyn seems to regulate neuronal toxicity and survival via the formation of channels or pores in the plasma membrane or via its interaction with other channels or receptors crucial for the proper neuronal activity (Figure 1).

### 2.5. Dopamine Metabolism

Soluble aSyn has been proposed to interact with the dopamine transporter (DAT) and decrease its amount on the plasma membrane, thus regulating the dopamine re-uptake from the synapse and protect neuronal cells from excessive dopamine toxicity [213,214]. Contrariwise, aSyn aggregation triggers DAT recruitment to the plasma membrane that results in massive entry of dopamine and production of reactive oxygen species (ROS) in neurons [215]. It is obvious that aSyn-mediated modulation of DAT activity is crucial for neuronal functioning via a balanced dopaminergic neurotransmission. Moreover, the regulation of dopamine storage is provided by an interaction of aSyn with the vesicular monoamine transporter 2 (VMAT2), which is responsible for the packaging of monoamine transmitters into synaptic vesicles [216]. It has been reported that increased levels of aSyn lead to VMAT2 inhibition and dopamine dysregulation that results in pathological events [217]. In addition, aSyn regulates dopamine biosynthesis, via reducing the activity or the phosphorylation status of TH, the rate-limiting enzyme in catecholamine synthesis [218,219,220,221,222,223]. In agreement, enhanced expression or phosphorylation and subsequent aggregation of aSyn alter TH activity and evoke an imbalance in dopamine synthesis, thus leading to neurotoxicity [132,224,225,226]. In vivo evidence further support the role of aSyn in dopamine metabolism, since the absence of aSyn caused decreased reuptake of dopamine, low levels of TH and DAT in the mouse striatum and reduced number of dopaminergic cells in the substantia nigra of aSyn KO mice [227,228,229].

### 2.6. Interaction with Mitochondria and ER

alpha-Synuclein displays a remarkable conformational flexibility upon macromolecular interactions and can associate with mitochondrial membranes, thus altering mitochondrial function [230,231,232] (Figure 2). There are reports suggesting that aSyn is a physiological regulator of mitochondrial activity [233,234,235], whereas others support the opposite [236,237,238]. Such discrepancies could be attributed to the different synuclein models utilized in each study, taking into account that brain homeostasis is a complex process and in vivo studies recapitulate better the interplay between the various brain components, compared to the isolated in vitro cellular setup. A bidirectional interaction between aSyn aggregation and mitochondrial dysfunction has been implicated in PD pathogenesis. In particular, increased levels of aSyn can lead to mitochondrial dysfunction [239,240,241,242,243,244], whereas, conversely, impairment of mitochondrial activity may accelerate aSyn pathology [245,246,247,248]; however, the precise underlying mechanisms remain to be elucidated. Both WT and mutant aSyn have been shown to interact with mitochondrial elements, altering both mitochondria morphology and function. Specifically, soluble pre-fibrillar aSyn oligomers seem to be responsible for complex I dysfunction, loss of membrane potential, disrupted Ca^2+^ homeostasis, enhanced cytochrome c release and ROS production, thus leading to neuronal demise [240,249,250,251,252].

Experiments in various animal models of a-synucleinopathy have revealed mitochondrial abnormalities, DNA damage and neuronal degeneration in PD-affected brain regions [244,253,254]. Moreover, in vitro and in vivo experiments have shown that aSyn inhibits mitochondrial fusion and triggers mitochondrial fragmentation [231,255]. Di Maio and colleagues have proposed that certain post-translationally modified aSyn conformations (soluble oligomers, dopamine-modified and S129E phosphorylation mimic) lead to impaired mitochondrial function via binding to TOM20 (translocase of the outer membrane receptor) and inhibiting mitochondrial protein import [239].

Nonetheless, there is evidence suggesting an impairment of mitochondrial function upstream of aSyn pathology. Experiments using the pesticides rotenone and paraquat have shown that dysregulation of mitochondrial function leads to nigrostriatal dopaminergic loss and formation of LB-like inclusions, positively stained with anti-aSyn antibodies and thioflavine S, thus resembling PD features [246,247,256,257,258]. Similarly, incubation of WT aSyn-overexpressing COS-7 cells with mitochondrial inhibitors resulted in the disappearance of the aSyn aggregates formed upon rotenone or oligomycin treatment [259]. A plethora of studies that utilize the mitochondrial neurotoxin MPTP to induce PD-like pathology in animals, further suggest that mitochondria impairment is a key player in disease development [245,248,260,261,262,263,264]. Genetic studies further support the hypothesis of aSyn accumulation as a secondary event following mitochondrial malfunction. Specifically, mutations in ATP13A2 (ATPase cation transporting 13A2), encoding for the lysosomal type 5 P-type ATPase, were shown to result in dysregulation in mitochondrial depolarization and ATP metabolism leading to mitochondrial fragmentation and subsequent cell death [265,266].

Apart from its implication in mitochondrial failure, aSyn has been also reported to play a biological role in the association of mitochondria with the endoplasmic reticulum (ER) Ca^2+^ homeostasis. It has been demonstrated that aSyn favors the Ca^2+^ transfer from ER to mitochondria, as a result of the communication the two organelles, probably due to the fact that aSyn can act as a “bridge” via its C terminus [267]. Later studies further supported the physiological localization of aSyn in mitochondria-associated ER membranes (MAM), stabilizing their interaction, which was perturbed upon aSyn aggregation and its subsequent redistribution [268,269]. Interestingly, the familial PD-linked A53T and A30P aSyn point mutations resulted in their weakened interaction with MAM, which affected MAM function and mitochondrial integrity [269].

The association of aSyn with mitochondria was further corroborated by findings indicating interactions between both monomeric and oligomeric aSyn with the Ca^2+^ transporting voltage-dependent anion channel 1 (VDAC1) [270,271,272,273]. Importantly, VDAC1 has been detected on the MAM of ER mediating the communication between the two organelles, regulating Ca^2+^ homeostasis [274,275,276]. Moreover, VDAC levels have been found decreased in nigral neurons of PD brains, where pathological aSyn inclusions had been formed [277]. Additionally, VDAC has been proposed to be a component of the mitochondrial permeability transition pore, the opening of which has been shown to be affected by aSyn overexpression and oligomerization [230,278]. In vivo experiments on transgenic mice overexpressing the human A53T aSyn further supported the role of permeability transition pore activity modulation on the mitochondrial dysfunction during PD pathogenesis [279].

### 2.7. Unfolded Protein Response, Regulation of ER/Golgi Trafficking and Ca^2+^ Homeostasis

The ER is a continuous membrane system mainly responsible for the production and processing of lipids and proteins, as well as Ca^2+^ homeostasis. In case of impaired protein folding (ER stress), cells activate a group of signal transduction pathways, known as the unfolded protein response (UPR). It has been previously shown that aSyn overexpression in PD patients leads to UPR and contributes to the molecular pathogenesis of the disease [280]. The ER chaperone glucose regulated protein 78 (GRP78/BIP) has a crucial role on ER stress regulation due to its ability to control the activation of transmembrane ER stress sensors (IRE1, PERK and ATF6) [281]. Disassociation of GRP78 from IRE1 and PERK results in stress signaling, finally leading to altered ER homeostasis [282]. aSyn associates with GRP78/BIP under physiological or pathological conditions, thus inducing UPR and leading to dopaminergic cell death [45,283]. Strikingly, Ser129 phosphorylated and aggregated aSyn was found in ER microsomes of A53T transgenic mice and more importantly, administration of the UPR inhibitor salubrinal, effectively attenuated disease manifestations in this PD-mouse model [284,285]. It is worth mentioning that GRP78/BiP levels were found elevated in DLB and PD brains in an aSyn burden-dependent manner [286]. In addition, the protein levels of various ER chaperones were found elevated in a-synucleinopathy models, co-localized with aSyn positive inclusions, suggesting that aggregated aSyn could potentially be implicated in UPR regulation in disease progression [284,287,288,289,290,291,292,293].

Proteins synthesized in the ER, are packaged into vesicles and directed to Golgi apparatus for subsequent modifications. One of the first pathological roles attributed to aSyn is the blockade of the vesicular transport from ER to Golgi by antagonizing ER/Golgi SNAREs [294,295,296]. Towards the same direction, aSyn can also disrupt the intra-Golgi and post-Golgi secretory trafficking, via an abnormal interaction with several Rab-family proteins of the intracellular endocytic pathway [294,296,297,298,299]. Additionally, aSyn can also impair the ionic transport and membrane trafficking, resulting in Golgi fragmentation and subsequent cytotoxicity [300,301,302].

Another significant role of aSyn on ER and Golgi function is the regulation of Ca^2+^ homeostasis via its binding on specific channels or pumps localized in these organelles (Figure 2). Specifically, proximity ligation assay experiments demonstrated that soluble and insoluble aSyn aggregates, but not monomers, interact with the ER Ca^2+^-ATPase SERCA, resulting in decreased cytosolic Ca^2+^ that disrupts the physiological cell function and leads to neuronal cell death [303]. Moreover, administration of the SERCA inhibitor cyclopiazonic acid restored cytosolic Ca^2+^ levels and protected neurons against the aggregated aSyn-dependent cell death [303]. In support to these results, aggregated aSyn bound on SERCA pump was detected in LBs and GCIs of PD and MSA brains, respectively [303]. Furthermore, PMR1, a Ca^2+^-transporting ATPase 1 pump regulating the levels of Ca^2+^ and Mn^+2^ ions in the Golgi [304], has been proposed to be a mediator of aSyn-dependent cytotoxicity. Specifically, in various PD models (yeast, flies and nematodes), PMR1 pump has been linked to aSyn pathology via a Ca^2+^-dependent mechanism, where aSyn accumulation elevated cytosolic Ca^2+^ levels and increased cell death. Interestingly, upon PMR1 deletion, the disease-associated characteristics were abolished, further suggesting the relevance of this pump to aSyn pathology [305,306].

### 2.8. a-Synuclein in the Nucleus

The name aSyn was given to the protein due to its localization in the nucleus and presynaptic nerve terminals [37]. Nuclear aSyn was detected in neurons of various brain regions of rodents and was reported to interact with histones, underlying PD pathology [307,308,309], even though a single study declares that the nuclear staining of aSyn is attributed to the non-specific signal of some antibodies that probably recognize unknown antigens in neuronal nuclei [310]. It has been proposed that aSyn is responsible for epigenetic dysregulation via inhibition of histone acetylation or reduced DNA methylation, thus favoring neuronal degeneration, whereas others suggest that nuclear aSyn regulates cell cycle rate exhibiting cell toxicity [311,312,313]. Importantly, histone deacetylase (HDAC) inhibitors attenuated aSyn toxicity and provided neuroprotection in both cell culture and transgenic Drosophila models [311,314].

Experiments in SH-SY5Y cells revealed that nuclear translocation of aSyn is regulated by calreticulin and Ca^2+^, following treatment with retinoic acid and modulates the expression of PD-linked genes such as ATP13A2 and PINK1 (PTEN-induced kinase1) [315]. Interestingly, phosphorylated aSyn at Ser129 was found accumulated in the nucleus of HEK293E-aSyn overexpressing cells and in various brain regions of transgenic (Thy1)-[A30P] aSyn mice [316]. Further experiments in H4 cells expressing various aSyn proteins verified that nuclear localization of aSyn depends on its phosphorylation at Ser129 [317]. The same group supported a role of DNA-binding and gene expression regulation for aSyn providing an insight into the role of modified aSyn in the nucleus [317]. Furthermore, other post-translational modifications of aSyn, such as sumoylation, seem to be responsible for the translocation of aSyn from the cytoplasm to the nucleus [318]. Although the majority of studies support a neurotoxic role for aSyn nuclear localization, some groups proposed that aSyn in the nucleus displays a protective role against DNA damage, replication stress or impaired nucleo-cytoplasmic transport [319,320,321]. However, the numerous in vitro and in vivo studies demonstrating a neurotoxic role of nuclear aSyn, in contradiction to the limited number of studies supporting a protective role originated mostly from cell lines or yeast, favors the pathological potential of nuclear aSyn.

### 2.9. Alpha-Synuclein and Protein Degradation Pathways: An Intricate Interplay

A great wealth of data focuses on the complicated relationship between aSyn clearance and protein degradation pathways (Figure 2). Both the ubiquitin-proteasome system (UPS) and the autophagy lysosome pathway (ALP) are responsible for aSyn degradation in a manner that depends on cell type, tissue and aSyn conformation state [322,323,324]. Specifically, there are studies demonstrating that aSyn can be degraded by the 26S/20S proteasome via ubiquitin-dependent [325,326] and ubiquitin-independent manner [327,328]. Studies in PC12, HEK293 and primary mesencephalic cells suggested that pharmacological inhibition of the proteasome does not lead to aSyn accumulation [324,329,330]; however, others have shown that soluble aSyn oligomers, but not monomers, are partially cleared via the 26S proteasome [331]. Importantly, it has been proposed that the UPS is responsible for aSyn removal under normal conditions, while in pathological cases the ALP is recruited to clear the increased aSyn burden [332].

Chaperone-mediated autophagy (CMA) is also responsible for the degradation of monomeric or dimeric forms of the protein via the lysosome-associated membrane protein type 2A (LAMP2A), whereas oligomeric aSyn is cleared mainly via macroautophagy [324,333,334]. Lee and colleagues also suggested that the lysosome is responsible for the removal of oligomeric but not fibrillar aSyn and that lysosomal failure results in aSyn accumulation and aggregation and subsequent cell death [335]. Moreover, initial in vivo evidence suggested that increased aSyn protein levels evoked by paraquat treatment were preferably degraded via CMA in dopaminergic neurons, where the levels of LAMP2A and the lysosomal heat shock cognate protein of 70 kDa (HSC70), both essential CMA-components, were found elevated [336]. We have also shown that boosting CMA function via LAMP2A overexpression in cell lines and primary neuronal cultures and in the rat dopaminergic system mitigated aSyn protein levels and related toxicity [337]. Similar neuroprotective effects were obtained upon LAMP2A overexpression in the Drosophila brain [338]. On the contrary, we have also shown that LAMP2A silencing led to endogenous aSyn accumulation in vitro [324] and in vivo [339] and in extensive neurodegeneration of the rat nigrostriatal axis [339]. Decreased levels of LAMP2A and HSC70 were reported in the human substantia nigra and amygdala of PD brains [340], whereas, in a subsequent study, LAMP2A was found to be selectively reduced in association with increased aSyn levels, even in the early stages of PD, thus suggesting a potential dysregulation of CMA-mediated protein degradation prior to substantial aSyn aggregation in PD [341].

However, a bidirectional link between aSyn accumulation and the protein degradation machineries exists and extensive studies have been conducted to elucidate not only the manner of aberrant aSyn degradation in a-synucleinopathies, but also the impact of various aSyn conformations on UPS and ALP function. It has been proposed that overexpression of A30P and A53T mutants, contrarily to WT aSyn, leads to cell death due to proteasomal inhibition [342]. Indeed, overexpression of mutant A53T aSyn resulted in UPS failure by inhibiting the activity of the 20S/26S proteasome, finally leading to aSyn pathological accumulation [343]. Other groups have failed to detect alterations in the proteasomal function of PC12 cells or transgenic mice, following overexpression of WT or mutant (A30P, A53T) aSyn [344]. Moreover, later studies demonstrated that transient overexpression of WT or mutant aSyn, followed by addition of recombinant aSyn oligomers and fibrils in an osteosarcoma cell line, did not result in any disturbance of the proteasomal function [345]. Importantly, studies in human post-mortem PD brains also suggested impaired proteasomal function in the substantia nigra [346,347,348], further supporting a role of UPS malfunction in PD pathogenesis. In addition, total rates of protein degradation declines with aging, thus contributing to the pathogenesis of age-related diseases [349]. Even though human post-mortem studies provide valuable information in regards to etiology and/or disease pathogenesis, the data obtained should be treated with caution, given into account the overall decline in the function of multiple systems with aging. For a-synucleinopathies, we believe that the use of tissue from affected and non-affected (in regards to aSyn pathology and neuronal death) brain areas may provide useful information regarding early or late events leading to neurodegeneration.

Increased aSyn protein burden is reported to impair macroautophagy function as well, via its interaction with Rab1a, an event that subsequently results in the autophagosome-formation-related protein Atg9 mislocalization [350]. Similar results were obtained from cells expressing the PD-linked mutation of the retromer protein VPS35, which is involved in autophagy and is implicated in PD pathogenesis [351]. The three most well studied PD-linked aSyn mutations, E46K, A30P and A53T, have been shown to promote ALP dysfunction, via either impairing autophagosome formation or inhibiting the selective removal of damaged mitochondria through mitophagy [352,353,354]. It has been previously reported that dopamine-modified aSyn inhibits CMA and this could probably shed light into the selective vulnerability of dopaminergic neurons in PD [355]. Further experiments in human iPSC-derived midbrain dopaminergic neurons revealed that disrupted hydrolase trafficking, due to aSyn overexpression, reduces lysosomal function [356]. Similarly, multiple studies suggest that there is a strong relationship between decreased β-glucocerebrosidase (GCase) activity and aSyn accumulation. In particular, heterozygote mutations in *GBA1* gene encoding for β-glucocerebrosidase represent a major risk factor for PD development with a-synucleinopathy [357,358,359,360,361,362].

### 2.10. Alpha-Synuclein in the Extracellular Space

The first indication that aSyn can be secreted arose from the detection of the protein in human CSF and plasma of PD patients, indicating that aSyn can be released into the extracellular space [363,364] and can exert various deleterious effects on neighboring cells. Further studies supported that aSyn can be secreted from neuronal cells, either via vesicles or exosomes [365,366,367]. Extracellular aSyn has been the subject of intensive research in recent years, mainly due to its propensity to spread from neuron to neuron or other glial cells, as discussed in the following sections.

The major hypothesis regarding the onset and spread of aSyn pathology in a-synucleinopathies relies in the protein’s nucleation propensity that leads to the formation of aberrant aSyn species, which then spread to neighboring cells and tissues via various mechanisms. Furthermore, aSyn has been proposed to act as a “prion-like” protein since it was demonstrated that pathogenic aSyn could transfer from diseased neurons of a PD patient to the healthy transplanted ones, fourteen years after the surgical intervention [368]. Similar results were obtained by other groups in both humans and rats [369,370,371,372]. Experiments of PD and DLB patient-derived brain extracts delivered into the brain of mice and non-human primates further validated the transfer of pathological aSyn and the formation of aSyn aggregates within the recipient neurons [373,374]. Moreover, when Pre-Formed Fibrils (PFFs) were used as seeds in both in vitro and in vivo experiments, the endogenous neuronal aSyn was recruited into the formation of highly insoluble aggregates [104,375,376,377,378,379].

Various mechanisms have been proposed for aSyn spread throughout the nervous system, following its release from neurons where the protein is normally expressed. Candidate mechanisms include aSyn secretion via vesicles, exosomes or even naked protein [364,365,366,380,381,382,383] and its uptake from the cells via conventional endocytosis [384,385], passive diffusion [386], tunneling nanotubes [387], membrane penetration [195,388,389] or receptor-mediated internalization [206,390,391]. Once taken-up by recipient cells, the exogenous aSyn has been shown to trigger the endogenous aSyn accumulation via an unknown mechanism [392,393,394,395]. However, according to the prevailing hypothesis, upon the cell-internalization of aberrant aSyn conformations (oligomers or fibrils), these serve as a template for the recruitment of the endogenous monomeric aSyn into the formation of insoluble aggregates [373,375,376,377,396,397,398]. The prevalently unfolded or alpha-helical aSyn is triggered to self-assemble generating fibrils that subsequently deposit as Lewy bodies [399,400,401].

Neuron-to-neuron aSyn transmission occurs following both anterograde and retrograde axonal transport or trans-synaptic pathways [402,403,404]. Several groups have proposed that dysregulation of axonal transport is implicated in aSyn accumulation at the cell body; however, it is not clear whether PD-linked aSyn mutations play a key role in the process per se [403,405,406,407]. Notably, aSyn in its oligomeric form has been shown to interfere with microtubules and kinesin motors, thus disrupting the anterograde transport and similar results were obtained in an aSyn overexpressing mouse model for PD, as well as in patients diagnosed with the disease [408,409,410]. Additionally, it has been suggested that the variety in a-synucleinopathy phenotypes is attributed to the formation of different aSyn “strains” that display “aggressive” characteristics [17,18,411]. As a consequence of their disparate structures, these “strains” have discrete biochemical responses along the different brain regions and cell types, thus explaining the various disease manifestations of a-synucleinopathies [19,20,125,412,413,414].

## 3. Glia in the CNS: Scavengers of Extracellular aSyn

### 3.1. Role in Microglia Function and Dysfunction

Microglial cells are the resident phagocytes of the brain, guarding the CNS homeostasis and performing essential role in health and disease. Specifically, apart from exhibiting immunoreactivity as a response to any changes or inflammatory stimulus, they are responsible for the monitoring and pruning of neuronal synapses [415,416,417,418,419]. Disturbance or loss of brain homeostasis “activates” microglial cells, a term used to describe the changes in their shape, gene expression profile and function during their response [420,421,422,423]. A well-regulated immune surveillance of the brain is essential for the proper CNS functioning; however, an excessive and continuous inflammatory response could potentially lead to cellular and tissue damage, tightly linked to the development of various neurodegenerative diseases [424,425,426,427,428]. Enhanced production of pro-inflammatory cytokines, reactive oxygen species (ROS, NO, superoxide) and glutamate has been shown to lead to dopaminergic cell death in PD [429,430,431].

Indeed, in the diseased brain, microglial cells get activated in two states, M1 and M2, depending on the cytokine signaling pathway involved. The classical pro-inflammatory TNF/IFNγ-mediated activation leads to M1 state, whereas the M2 state is subdivided into the M2a “alternative activation” and the M2c “acquired deactivation” states, acquired following the involvement of anti-inflammatory cytokines IL-4 and IL-13 (for M2a) or IL-10 and TGF-β (for M2c) [432]. It has been proposed that in PD a shift from M2 to M1 phenotype is responsible for disease progression; therefore, the scientific interest has been focused on immunomodulatory therapies promoting the neuroprotective M2 type [433].

#### 3.1.1. Physiological Role of aSyn in Microglia Function

Although aSyn has been primarily characterized as a pre-synaptic neuronal protein, several studies have proposed a physiological role of aSyn in microglial function. Microglial cells from mice lacking aSyn (aSyn -/-) displayed reduced phagocytic activity and enhanced secretion of pro-inflammatory cytokines, thus suggesting that aSyn modulates the activation phenotype of the brain immune cells and contributes to the clearance of debris present in the local brain microenvironment [434,435]. On the other hand, transient overexpression of WT, A53T or A30P aSyn in BV2 microglial cells led to a distinct pro-inflammatory cytokine profile in combination with impaired phagocytic activity [436]. Additionally, microglia isolated from BAC transgenic mice overexpressing aSyn, exhibited dysregulation of cytokine release and phagocytosis [437]. Data obtained from iPSC-derived macrophages from PD patients harboring the A53T aSyn mutation and aSyn triplication mutations further support the implication of aSyn in the phagocytic capability of these cells [438]. Expression of aSyn in microglia has been also proposed to promote cell migration via the enhanced expression of the cell-surface glycoprotein CD44 and the matrix metalloproteinase membrane-type 1 (MMP-MT1) [439]. However, the presence of various aSyn species in the environment of microglial cells alters their physiology and behavior leading to neuroinflammation and neurodegeneration.

#### 3.1.2. Microgliosis in a-Synucleinopathies

Microgliosis is the reaction of CNS microglial cells to pathogenic insults and their shift from a resting to the active state [440]. Since the first study demonstrating microgliosis in PD brains [25], a plethora of reports highlights an important role of activated microglia in disease pathogenesis in both humans and animal models (reviewed in [441,442,443,444,445]). Microglial activation has been observed in PD brains by in vivo positron emission tomography (PET) imaging studies [446,447,448], suggesting that microgliosis is an early event that perpetuates during the disease progression. Additional studies have further supported the hypothesis of an early activation of microglia tightly associated to aSyn pathology, in various PD models [192,449,450,451,452,453,454]. However, other reports suggested that microglia respond differently in the various disease stages, in a manner that depends on the affected brain region and the protein burden of aSyn, indicating the existence of immunological diversity among microglia in the diseased brain [449,450,455].

It has been proposed that the neuron-microglia interaction may contribute to the neuroinflammation that characterizes PD, where neurons expressing aSyn activate microglia, which in turn secrete inflammatory factors surrounding the diseased neurons, thus forming a vicious cycle [456] (Figure 3). Likewise, microgliosis has been also reported in MSA, where aSyn is found aggregated mainly within oligodendrocytes [426,457,458,459]. This hypothesis for neuron-microglia communication in synucleinopathies is further supported by findings demonstrating an altered expression profile of various cytokines in the brains of PD patients [460]. Moreover, microglial activation has been shown to induce aSyn-mediated neuronal cell death, in both in vitro and in vivo PD models [461,462,463]. In addition, microglial cells exposed either to cytokines or to PD-derived CSF, displayed alterations in the intracellular aSyn protein levels, suggesting a crucial role of the brain microenvironment for aSyn accumulation in microglia [464,465]. Activated microglia have been also detected in various transgenic animal models overexpressing wild type of mutated aSyn specifically in neurons [192,452,453,454,466,467]. Experimental PD animal models including the use of the MPTP mouse model or nigral injections of recombinant aSyn fibrils or AAVs overexpressing aSyn are characterized by neuroinflammation followed by a significant degeneration of dopaminergic neurons [449,450,468,469,470]. Similarly, further results obtained from the rAAV-driven overexpression of aSyn in the mouse substantia nigra show extensive aSyn-mediated microgliosis primarily in the nigrostriatal axis, accompanied by an increase in the production of pro-inflammatory cytokines [410,471,472].

#### 3.1.3. Activation of Microglia and Clearance of Toxic aSyn Species

Numerous in vitro and in vivo studies have demonstrated that either conditioned medium from aSyn overexpressing cells or aSyn per se (i.e., recombinant monomeric, oligomeric or fibrillar aSyn) robustly activate microglia. In particular, treatment of microglial cells with non-aggregated aSyn was shown to increase phagocytosis and enhance pro-inflammatory cytokines release, NF-kB nuclear translocation and microglial migration [439,468,473,474,475], whereas addition of fibrillar aSyn in BV2 cells was reported to reduce their phagocytic activity [474]. In agreement, incubation of human microglial cell lines or primary microglial cells with monomeric aSyn triggered the release of various pro-inflammatory factors [476,477,478,479]; however, recently, it has been suggested that monomeric, contrarily to oligomeric aSyn, promotes an anti-inflammatory phenotype of microglia [480]. Other groups have found that aggregated aSyn leads to increased TNFa and ROS production, both related to cell toxicity [456,481,482,483].

Microglial activation may depend on the aggregation state of aSyn and microglial cells readily take-up fibrillar aSyn and produce pro-inflammatory cytokines [484]. Furthermore, incubation of microglial cells with conditioned media from neuronal cells or with CSF from PD patients resulted in significant secretion of TNFa, IL1β and ROS [465,473,478]. Importantly, it has been shown that PD-related aSyn mutants are more prone to activate microglia when compared to WT protein [463,479,485]. Moreover, elevated levels of CXCL12 chemokine in both postmortem PD brain tissue and in nigral microglia of transgenic A53T mice further support the aSyn-mediated neuroinflammation [486]. Contrariwise, aSyn-evoked microgliosis in some instances leads to the enhanced expression and release of neuroprotective factors, such as BDNF, probably in an attempt for neuronal repair and survival [481]. Notably, studies in microglia of mice lacking aSyn expression have verified the critical role of aSyn in modulating microglial activation [434,435].

aSyn has been proposed to enter microglial cells via a clathrin-mediated endocytosis and leads to neurotoxicity in a Prostaglandin E2 receptor subtype 2 (EP2)-dependent manner [470,487]. Integrin CD11b has been proposed as a mediator of aSyn-induced activation of microglial cells [488,489,490], whereas other membrane receptors implicated in microglial stimulation in the presence of aSyn are the protease activated receptor 1 (PAR-1), the macrophage antigen 1-receptor (Mac-1), the Fcγ receptors (FcγR), the microglial purinergic receptor P2X7 and the scavenger receptor CD36, as well as the adhesion molecule CD44 and some plasma membrane ion channels [439,463,477,478,483,491,492,493]. Importantly, Toll-like-receptors 2 and 4 (TLR2 and TLR4) are considered crucial modulators of glial responses in a-synucleinopathies, as well as key players in aSyn phagocytosis. These properties have put them in the spotlight as novel therapeutic modulators of neuroinflammation in neurodegenerative diseases [494,495,496,497,498,499,500,501,502].

The internalization of aSyn by microglia triggers various immune response-related cascades, including NF-kB, Nrf2, MHCII and inflammasome. Numerous studies have reported activation of NF-kB pathway upon addition of various aSyn conformations in both rodent and human microglial cell lines [468,469,476,478,485,503]. The nuclear translocation of NF-kB is a result of aSyn interaction with TLR that leads to the Myd 88-mediated activation of IkB kinases [504]. Moreover, oligomeric aSyn has been shown to trigger TLR2 signaling in microglial cells via NF-kB and p38 MAPK activation, which has been previously linked to aSyn-related toxicity [469,505]. Another key player in both aSyn pathogenesis and the neuroimmune system is LRRK2 (Leucine-rich repeat kinase 2), constitutively expressed in neurons and glial cells, mutations of which have been characterized as common risk factors for PD. Significantly, manipulation of LRRK2 expression levels in mouse microglial cells has revealed its role in regulating aSyn degradation [506,507].

Furthermore, various studies have proposed that aggregated aSyn results in IL-1β production by reactive microglia, which in turn involves nod-like receptor protein 3 (NLRP3) inflammasome activation [508,509,510]. Interestingly, the inflammasome related caspase-1 activation is responsible for aSyn truncation and its subsequent pathological accumulation [188]. Apart from the activation of microglial pro-inflammatory transcription factors, such as NFkB, pathological aSyn also affects the antioxidant transcription factor Nrf2 [485,511]. Modifications in the expression levels of Nrf2 have confirmed its role in aSyn clearance and neuronal survival [512,513,514]. Moreover, since microglial cells act as antigen presenting cells in the brain, it has been proposed that upon aSyn internalization by microglia, the protein is presented to T-cells via MHCII, which then infiltrate in the CNS and finally lead to neuronal degeneration present in PD and MSA [22,515,516,517,518,519,520].

Apart from modulating the immune responses in the CNS, microglial cells are the brain’s phagocyting cells clearing cellular debris or any toxic insult. Amongst the various threats, extracellular aSyn has been shown to be effectively processed by activated microglia, in some cases via a DJ1-mediated autophagy [474,521,522,523,524]. Consistent with these results, experiments utilizing primary cells, have demonstrated that aSyn upon its internalization by microglia is targeted to autophagosomes, thus leading to its degradation [493]. TLR4 also seems to play a crucial role in microglial phagocytosis, since TLR4 (-/-) murine microglial cells exhibit impaired aSyn clearance and enhanced neurotoxicity [494,496]. Furthermore, addition of aSyn fibrils in BV2 and primary microglial cells induces autophagy as a rescue mechanism to restore lysosomal damage [525].

Importantly, aging is another key player in the efficient aSyn clearance, since it has been proposed that microglia and monocytes display reduced phagocytic activity with age [526,527]. Regarding the uptake of aSyn by microglia, it has been reported that ganglioside GM1 and lipid rafts, but not clathrin, caveolae and dynamin, mediate monomeric aSyn internalization, whereas aggregated aSyn enters microglia via clathrin- and calnexin-dependent mechanisms [487,528]. Moreover, microglial uptake of neuronally-derived exosome-associated aSyn via macropinocytosis could potentially account for pathological aSyn spreading [529,530,531]. Strikingly, apart from the immunomodulating role of microglia in the CNS, it has been also suggested that resting microglia regulates the cell-to-cell transfer of aSyn in vivo [532]. Therefore, further considering the aforementioned involvement of exosomes in the transmission of aSyn pathology, targeting exosome-release from various cell types of the brain could be a potential therapeutic target against disease progression.

### 3.2. Astrocytes in a-Synucleinopathies

Astrocytes, the star-shaped cells of the brain, are the most abundant glial cells of the CNS, accounting for at least one third of the brain mass. They have a supportive role to neurons, by maintaining osmotic, energetic and structural tissue homeostasis. In particular, it is suggested that astrocytes regulate neurotransmitter removal from the extracellular space, synaptic transmission, myelination, brain energy metabolism and pH homeostasis, ion balance, cholesterol synthesis, blood–brain barrier (BBB) permeability, cerebral blood flow and glymphatic system function [533,534,535,536,537,538,539,540,541,542,543]. It has also been suggested that astrocytes modulate neuronal synaptic activity via a Ca^2+^-dependent release of transmitters that have a feedback action on neurons, a process known as “gliotransmission” [544,545,546]. There are two main types of astrocytes in the brain: protoplasmic and fibrous [547]. Protoplasmic astrocytes are highly branched and are mainly located in the gray matter, tightly associated with neuronal cell bodies and synapses [548], whereas fibrous astrocytes have straight and long processes and they are widely distributed in the white matter, contacting nodes of Ranvier [549].

Apart from their crucial role in brain homeostasis and health, astrocytes have been also implicated in the cascade of events underlying neurodegenerative diseases. Specifically, astrocytes in PD brains have been reported to migrate and become reactive and have been classified in two categories depending on their neuroprotective or neurotoxic effects: harmful A1 astrocytes and protective A2 astrocytes [24]. Activated microglial cells induce A1-astrocyte reactivity, leading to neuronal and oligodendroglial cell death and subsequent synaptic impairment [24]. They have been also shown to secrete inflammatory cytokines, partially as a response to the increased aSyn protein load, thus contributing to PD progression and neurodegeneration [550], although there are studies reporting a neuroprotective role of astrocytes in a-synucleinopathies, modulating the levels of accumulated aSyn and protecting neurons against oxidative stress [551,552].

#### 3.2.1. Astrocytes in PD: Friend or Foe?

Up to date, there is little evidence regarding the expression levels [553] and the physiological role of aSyn in astrocytes, such as the implication of aSyn in astrocytic fatty acid metabolism [90]. It is also suggested that cultured human astrocytes express low levels of aSyn and various inflammatory cytokines or cell stress enhance aSyn production [554,555]. Yet, numerous studies have focused on the role of astrocytes in the modulation of aSyn levels and the regulation of immune responses in neurodegeneration. Interestingly, aSyn-positive inclusions have been detected in astrocytes in various regions of PD and DLB brains [13,14,15,556,557,558,559,560,561]. According to the prevailing hypothesis, the accumulated aSyn detected in astrocytes origins from the neighboring diseased neurons, which upon its release is subsequently internalized by astrocytes, probably as a mechanism of clearance and neuroprotection [562] (Figure 3). However, the responses of astrocytes in aSyn-related pathogenesis have been characterized as a “double-edged sword”, due to the controversial findings regarding their role in neurotoxicity or neuroprotection [563].

The neuron-to-astrocyte aSyn transmission has been extensively studied and is considered as the prime suspect for the detection of aSyn immunoreactivity within astrocytes in a-synucleinopathies. A seminal study utilizing primary astroglial cultures treated with conditioned media from differentiated SH-SY5Y neuronal cells showed that internalization of neuronally-derived aSyn by astrocytes occurs via endocytosis and results in the formation of proteinase-K resistant aSyn inclusions within astrocytes [550]. Similar results were obtained in transgenic mice overexpressing human aSyn under the neuronal promoter PDGFβ, where aSyn-positive inclusion bodies were observed in both neurons and astrocytes [550]. Furthermore, co-culture of primary astrocytes with SH-SY5Y cells verified the formation of LB-like inclusions positive for aSyn staining [564]. The detection of aSyn within the astrocytic endosomal/lysosomal compartment led to the hypothesis that astrocytes normally internalize neuronally-derived aSyn as a neuroprotective mechanism; however, prolonged exposure to pathological aSyn species may lead to impaired glial lysosomal function and, thus, astrocytic aSyn accumulation [550,565]. In agreement, overexpression of PD-linked aSyn mutants (A30P and A53T) in actrocyte cell lines resulted in impaired autophagic function, mitochondrial failure and cellular apoptosis, thus preventing the astrocyte-mediated neuroprotection [566]. In support to a protective role of astrocytes during PD, it has been reported that neurotrophin-immunoreactive (BDNF and NT-3) astrocytes surrounding degenerating nigral neurons in the brains of idiopathic PD patients may indicate a beneficial role of glial cells against neuronal failure [567]. In addition, elevated levels of the glutathione peroxidase-GPx in astrocytes of the substantia nigra of PD patients, further demonstrates their role against neuropathology [568,569]. Strikingly, overexpression of the Nrf2 transcription factor selectively in astrocytes in the haSyn A53T transgenic mouse model provided protection against aSyn-related toxicity by enhancing its degradation via the ALP [513]. In a recently published study, it was suggested that astrocytes internalize aSyn aggregates and effectively degrade them via proteasomal and autophagic pathways, thus protecting dopaminergic neurons against the aberrant effects of toxic aSyn species [570].

In vitro and ex vivo studies proposed that aSyn can efficiently transmit not only from neurons to astrocytes, but also between astrocytes and translocate to the lysosomes of the recipient cells [571]. However, differently from neurons, astrocytes are able to efficiently degrade fibrillar aSyn, suggesting an active role for these cells in clearing aSyn deposits [571]. It is worth-mentioning that the transfer of aSyn from astrocytes to neurons was reported to a lesser extent, even though another study suggested that aSyn deriving from SNCA-flag tagged PD astrocytes was effectively transferred to the co-cultured surrounding control neurons, which then displayed signs of degeneration [572]. Strikingly, iPSC-derived astrocytes from PD neurons displayed impaired CMA and macroautophagy, which could potentially account for the observed aSyn accumulation, highlighting the implication of astroglial-mediated proteolysis in the spread of aSyn pathology [572]. On the contrary, when astrocytes derived from healthy controls were co-cultured with PD neurons, the former cells absorbed the neuronal aSyn from the medium, thus indicating a potential neuroprotective effect of astrocytes [572].

In another aSyn-overexpressing neuron-astrocyte co-culture, addition of aSyn fibrils led to the formation of aSyn aggregates in both cell types, probably recapitulating the events occurring during PD pathogenesis [573]. It has been also suggested that aSyn enters astrocytes more efficiently than neurons via endocytosis and that aSyn transmission from astroglial to neuronal cells leads to neurotoxicity and cell death [574]. Indeed, the uptake of aSyn by astrocytes has been reported as an early-in-time event, since it was detected in the cytoplasm of the cells within 30 min following aSyn application [574]. The spread of aSyn from neurons to astroglial cells has also been demonstrated in rodent transgenic animal models overexpressing human aSyn in neuronal cells, by the detection of aSyn-positive inclusions within the cytoplasm of these glial cells [575,576]. Furthermore, hippocampal delivery of aSyn fibrils in the brain of M83 A53T Tg mice resulted in the formation of pSer129-positive aSyn inclusions within both astrocytes and microglia, four months post-injection [397],

Various mechanisms have been proposed to underlie aSyn transfer between neurons and astrocytes. For example, it is known that astrocytes, upon oxidative stress, form tunneling nanotubes (TNTs) in order to connect with other non-stressed cells [577]. It has been proposed that various stressors, such as aberrant aSyn, lysosomal dysfunction or mitochondrial failure could trigger TNT formation in astrocytes and enhance aSyn spreading [387,578]. Another mechanism for intercellular communication is via exosomes. Although extensive work has been done regarding the role of exosomes in neuronal aSyn transmission and disease pathology [366,579,580,581,582], few studies have proposed a vesicular-mediated transfer of neuroprotective molecules from astrocytes to neurons [583,584] or an exosome-related aSyn spread from neurons to astrocytes [585,586].

Upon aSyn transmission to astrocytes, the latter produce multiple pro-inflammatory cytokines (IL-1α, IL-1β, IL-6, IL-18) and chemokines (CC-, CXC- and CXCL-type) as a response [550,587]. It has been suggested that the pro-inflammatory response of astrocytes to aSyn depends on TLR4 [496,588]. Apart from cytokine release, aSyn leads to Ca^2+^ flux and oxidative stress upon its entry in astroglial cells, presumably leading to neurotoxicity [589,590,591]. Moreover, in vitro and in vivo experiments have shown that pathological aSyn triggers microglial activation, followed by the induction of reactive A1 astrocytes, finally leading to neurodegeneration [592]. It has been also reported that astrocytes over-expressing hA53T aSyn displayed impaired functions, including glutamate uptake and BBB regulation, resulting in paralysis in transgenic mice [593]. Additionally, aSyn-treated astrocytes have been proposed to produce reduced levels of cholesterol, whereas in parallel they display enhanced GFAP expression, indicative of astrocyte reactivity [594]. Significantly, treatment of astrocytes with various aSyn species (monomeric, oligomeric, fibrillar) induced astrocyte activation and secretion of TNF-α and IL-1β, the expression of which seemed to depend on aSyn species, leading to subsequent non-cell autonomous neuronal degeneration [595]. Other studies revealed that elevated expression levels of myeloperoxidase and enhanced IFN-γ signaling could mediate the astrocytic-activation and inflammation observed in PD brains [21,596,597,598].

alpha-Synuclein has been reported to trigger the opening of connexin 43 (Cx43) hemichannels and pannexin-1 (Panx1) channels in mouse cortical astrocytes, leading to alterations in [Ca^2+^]_i_ levels, production of nitric oxide (NO), enhanced purinergic and glutamatergic signaling, altered mitochondrial morphology and reduced astrocyte survival [599]. Another recently published study suggests that vesicle-associated aSyn, deriving from erythrocytes, effectively crosses the BBB and accumulates within astrocytes, impairing glutamate uptake, probably due to interactions of oligomeric aSyn with excitatory amino acid transporter 2 (EAAT2) [600]. Furthermore, astrocytes overexpressing mutant A35T and A30P aSyn triggered ER stress and damaged the Golgi apparatus, finally leading to apoptotic cell death [601]. Remarkably, co-culture of primary astrocytes overexpressing mutant aSyn with neuronal cells, inhibited neurite outgrowth, probably due to reduced GDNF secretion [601]. Finally, Cy3-labeled aSyn oligomers were internalized by glial cells, primarily astrocytes, which then started to degrade the oligomers via the ALP [602].

In addition to the well-established role of microglia in the activation of astrocytes, astrocytes themselves have been also reported to control microglial activation and microglia-induced neuroinflammation [603,604], thus unraveling an astrocyte-microglia intimate crosstalk (Figure 4). For example, in response to pathological aSyn insult, astrocytes can acquire a pro-inflammatory phenotype that can lead to neuronal death, independent of microglia. Given that astrocytes produce pro-inflammatory cytokines and chemokines as a response to various stimuli, it has been proposed that such astrocytes may mediate the microglial activation detected in aSyn-related brain diseases [605,606]. Various WT or mutant aSyn conformations have been shown to trigger the up-regulation of pro-inflammatory modulators in astrocytes, such as ICAM-1, IL-6 and TNF-α, leading to microglial activation, neuroinflammation and neurotoxic events during PD progression [496,587,607]. Specifically, transgenic mice inducibly overexpressing the PD-related A53T mutant aSyn selectively in astrocytes exhibited reactive astrogliosis accompanied by increased inflammatory responses and microglial activation in brain regions with significant neuronal loss [593]. Moreover, the detection of ICAM-1 positive reactive astrocytes surrounding brain areas with severe neuronal loss in PD brains or in the MPTP mouse PD model, indicates a sustained inflammatory process mainly triggered by astroglial cells, which is responsible for the consequent degeneration of dopaminergic neurons [608]. Finally, similar to the aSyn prion-like mode of action, the neurotoxic reactive astrocyte polarization has been recently proposed to occur during the CNS prion disease, where microglial cells seem to retain a neuroprotective phenotype against the inflammatory astrocytic responses [609]. All the above observations highly cement the contribution of astroglial reactivity in the pathogenesis of PD and related neuronal a-synucleinopathies.

#### 3.2.2. Implication of astrocytes in MSA pathology

Apart from PD and DLB, astroglial activation is also present in MSA and it seems to play a role in both disease initiation and progression. Extensive astrogliosis has been reported in various brain regions of MSA patients, in some cases accompanied by aSyn-positive inclusions within astrocytes [16,610,611,612,613]. In agreement, accumulation of abnormally phosphorylated and aggregated aSyn was present within astrocytes of MSA patients [614]. Interestingly, aSyn-positive doughnut-shaped inclusions were detected in radial processes of Bergmann glia (unipolar protoplasmic astrocytes in the cerebellar cortex) of various a-synucleinopathy brains, including MSA [558]. Although astroglial aSyn accumulation has been demonstrated in the brains of various MSA transgenic mouse models, finally leading to astrogliosis that accompanies neurodegeneration [615,616,617,618], Song et al. suggested that subcortical astrocytes in MSA did not display aSyn accumulation, in contrast to PD [14]. However, up-to-date the data on the precise role of astrocytes in MSA are scarce, reinforcing the necessity for further studies to elucidate the contribution of astroglial activation in MSA pathogenesis.

### 3.3. Alpha-Synuclein in Oligodendrocytes: The Pathologic Hallmark of MSA, A Unique Oligodendrogliopathy

Oligodendrocytes are responsible for the production of myelin [619] that surrounds the neuronal processes, mediates the transmission of electric signals between neurons and provides a neurotrophic support [620,621,622,623]. There are two main types of oligodendrocytes, the myelinating and the non-myelinating ones, that concentrate in white and grey matter, respectively [624]. Many neurodegenerative diseases occur due to either oligodendroglial death or damage to the myelin sheathes they produce leading to subsequent neuronal demise [622,625,626]. Oligodendrocytes have been proposed to participate in late PD and DLB progression, rather than in disease initiation [442,627]. This secondary involvement of oligodendrocytes in neuronal a-synucleinopathies was further supported by the detection of aSyn-positive inclusions within non-myelinating oligodendrocytes of PD and DLB brains [15,628], as well as by the presence of complement-activated oligodendrocytes in the diseased brains [629,630]. Moreover, axonal myelination deficits in neurodegenerative diseases also highlight the involvement of oligodendrocytes in neuropathogenetic events [631,632]. Additionally, oligodendroglial cell loss was reported in the striatum of the MPTP-intoxicated mouse PD model, shortly after MPTP administration [633].

On the other hand, the oligodendroglial aSyn inclusions detected in the brains of MSA patients (GCIs) are the main hallmark of the disease and are considered to play a crucial role in the primary events leading to MSA [626]. The involvement of oligodendrocytes in MSA initiation and progression is indisputable, due to the wide distribution of GCIs along the affected brain areas of MSA patients [611,634,635,636,637]. Apart from GCIs present within oligodendrocytes, other inclusions, such as neuronal cytoplasmic inclusions (NCIs) have been detected in neuronal somata, axons and nucleus in various brain regions, mainly composed of aSyn [638,639,640]. GCIs were first described three decades ago, as multi-shaped oligodendroglial inclusions composed of a 10-nm-sized central core fibrils, positively stained with antibodies against aSyn, surrounded by other aggregated proteins such as αβ-crystallin, ubiquitin, cytoskeletal proteins, chaperones and the microtubule-related proteins TPPP/p25α and tau [11,634,641,642,643]. Comparative analysis of the protein composition of GCIs and LBs revealed that GCIs consist of 11.7% aSyn, 1.9% αβ-crystallin and 2.3% 14-3-3 proteins, whereas LBs are composed of 8.5% aSyn, 2% αβ-crystallin and 1.5% 14-3-3 proteins [644]. It has been also reported that aSyn is a key player in the accumulation of tau and αβ-crystallin within GCIs [645], further highlighting the contribution of aSyn in disease pathogenesis.

Another early hallmark of oligodendroglial pathology in MSA and major component of GCIs is the oligodendroglial-specific phosphoprotein TPPP/p25α, which under physiological conditions has been proposed to mediate the myelination process and colocalize with myelin basic protein (MBP) in normal human brains [646,647,648]. Under pathological conditions, TPPP/p25α is considered to re-locate from the myelin sheaths to the abnormally expanded oligodendroglial somata and to trigger aSyn aggregation in vitro [12,649]. In vitro experiments utilizing TPPP/p25α ectopic overexpression in PC12 cells revealed that TPPP/p25α prevents the fusion of autophagosomes with lysosomes and impairs aSyn degradation, enhancing its secretion via exophagy [650]. Moreover, concurrent overexpression of TPPP/p25α and aSyn in OLN-93 rat oligodendroglial cells led to pSer129 aSyn-dependent microtubule retraction from the processes to the perinuclear space, as well as to cytotoxicity and subsequent cell death via activation of the FAS receptor and caspase-8 [651,652]. Recently published work from our lab revealed a crucial role for TPPP/p25α in the recruitment and seeding of oligodendroglial aSyn and in the formation of aberrant aSyn species within oligodendrocytes [653]. Additionally, the levels of glial cell-derived neurotrophic factor (GDNF) were found significantly decreased in the brains of MBP-haSyn transgenic mice, a mouse model for MSA where human aSyn is specifically overexpressed in oligodendrocytes [654]. Similar results were obtained from brain samples of MSA patients, further supporting that oligodendroglial aSyn accumulation is implicated in the dysregulation of neurotrophic support, oxidative stress and neuroinflammation, thus leading to MSA pathogenesis [654].

#### Alpha-Synuclein Accumulation in Oligodendrocytes, Propagation and Spread of Pathology

The origin of aSyn detected in oligodendroglial GCIs still remains enigmatic and there are controversial studies in the literature suggesting either the internalization of neuronally-secreted aSyn by oligodendrocytes or an enhanced expression and decreased degradation of oligodendroglial aSyn [655,656,657,658,659,660,661]. The release of aSyn by neuronal cells, partially bound on vesicles or exosomes is well-documented [365,366,380,662,663] and some studies propose that oligodendrocytes can take-up the neuronally-derived aSyn via dynamin GTPase-, clathrin- and dynasore-dependent mechanisms [658,664,665,666]. The neuron-oligodendrocyte communication can also be mediated via exosomes [667], which are characterized as “Trojan horses” of neurodegeneration [668] and they could serve as transporters of pathological disease-related proteins, such as aSyn (Figure 3). Moreover, ectopic expression of the endocytosis regulatory proteins Rab5 and Rabaptin-5 in oligodendrocytes of MSA brains may account for the elevated levels of aSyn within oligodendrocytes, probably via enhanced endocytotic activity [669].

In vitro and in vivo experiments revealed the ability of oligodendrocytes to take up exogenously added recombinant or neuronally-derived aSyn and incorporate it into intracellular GCI-like aggregates [658,660,664,665,666]. In a recently published study, mature human oligodendrocytes generated from neural stem cells had the ability to internalize neuronally-derived aSyn and form proteinaceous inclusions, thus further supporting the existing theory for the origin of MSA-related aSyn. Significantly, it has been shown that once neuronal aSyn is taken up by oligodendrocytes, it accumulates and gains GCI-like characteristics rather determined by the oligodendroglial milieu [17].

Over the last years, the prion hypothesis has gained a lot of attention regarding the spread of pathological aSyn in the context of both PD and MSA. Specifically, it has been reported that inoculation of transgenic mice overexpressing human A53T aSyn with MSA brain homogenates resulted in CNS dysfunction, whereas, strikingly, the PD brain-derived material did not evoke similar effects [670,671,672]. Similarly, intrastriatal injections of MSA homogenates in the brains of Tg(SNCA)1Nbm/J mice (out for mouse aSyn and overexpressing the human protein) resulted in the detection of hyper-phosphorylated aSyn-positive inclusions in various brain regions [673]. Finally, treatment of HEK293T cells stably expressing fluorescently-tagged aSyn with healthy, PD or MSA brain-derived extracts highlighted that only the MSA-added material was capable of inducing aSyn aggregation [674].

There are other possible scenarios that have been proposed to explain aSyn seeding and propagation in MSA brains, based on oligodendrocyte-to-oligodendrocyte communications. Specifically, it is possible that aSyn is taken up by oligodendroglial progenitor cells prior to their maturation, probably impairing the myelination process [675]. Finally, another scenario suggests that, in MSA pathology, oligodendrocytes adopt unknown cellular uptake mechanisms for aSyn internalization and subsequent propagation [675]; however, the precise mechanisms underlying aSyn transfer to oligodendrocytes still remains unknown. The gap junction protein connexin-32 (Cx32) has been also implicated in the uptake of oligomeric aSyn by both neurons and oligodendrocytes [676] and Cx32 protein levels were found elevated in animal PD and MSA models, thus suggesting an interaction between Cx32 expression and aSyn cellular uptake [676]. Contrariwise, others and we have suggested that the endogenous aSyn, expressed even at minute amounts, plays a pivotal role in the accumulation of pathological aSyn within oligodendrocytes and the subsequent GCI-like formation [653,677].

It is interesting to note that oligodendroglial and neuronal aSyn accumulation has been reported to occur in different time points and in particular that aSyn aggregation requires several months to progress within oligodendrocytes, upon synthetic haSyn-PFFs delivery into the brain of WT mice [678]. Moreover, in vitro aSyn overexpression in oligodendrocytes resulted in delayed maturation of oligodendrocyte progenitor cells and impaired myelin-gene expression and myelination deficits [679,680,681], whereas in another study aSyn-positive inclusions were mainly detected in BCAS1-expressing (breast carcinoma amplified sequence 1) immature oligodendrocytes of MSA brains [682]. The above observations insinuate that the oligodendroglial maturation and aSyn-aggregate formation are closely linked and may provide information regarding pathogenic events in MSA.

Regarding the hypothesis of impaired aSyn degradation in the context of MSA, both the UPS and the ALP have been proposed to contribute to the accumulation and aggregation of aSyn within oligodendrocytes. The detection of LC3-positive signal or other autophagy-related proteins, such as ubiquitin and p62 in GCIs points a role of the ALP in MSA pathogenesis [659,683,684,685,686,687]. It has also been suggested that AMBRA1, an upstream protein regulator of autophagy and UCH-L1, a deubiquitylating enzyme, are implicated in neurodegenerative diseases with oligodendroglia pathology [688,689]. The role of autophagic dysregulation along with mitochondrial impairment in aSyn aggregation was also studied in primary oligodendroglial cultures and in the OLN-t40 oligodendroglial cell line [690]. Moreover, neurosin (kallikrein 6) has been proven an effective serine protease in clearing aSyn from oligodendrocytes both in vitro and in vivo [691,692,693]. In addition, treatment of Tg haSyn-PLP mice, a well-established MSA mouse model, with the proteasome inhibitor I for 12 weeks, resulted in enhanced accumulation of both human and endogenous mouse aSyn within the cytoplasm of oligodendrocytes, thus highlighting the role of UPS in aSyn degradation [694]. Finally, several in vitro studies have proposed that aSyn aggregation is stimulated by heparin and heparan sulfate [391,695,696], linear polysaccharides (glycosaminoglycans) found on the cell membrane and in the extracellular matrix [697,698]. Heparan sulfate has been suggested to mediate aSyn fibril uptake by oligodendrocytes via binding to the plasma membrane [391,699], whereas other have proposed that heparin and heparan sulfate proteoglycans (HSPGs) are responsible for aSyn fibrillation [695,696,700,701,702].

Apart from aSyn toxicity *per se*, the overexpression of aSyn in oligodendrocytes can lead to oligodendroglial cell death and subsequent neuronal loss via a plethora of mechanisms. Specifically, aSyn-overexpressing oligodendrocytes are more susceptible to oxidative stress and various cytokine actions [591,703], or display impaired adhesion properties [704]. Furthermore, it has been reported that animal MSA models exhibit myelin loss and impaired mitochondrial function, accompanied by severe neurodegeneration in various brain regions [615,617,618,705,706].

## 4. Conclusions

It has been almost 25 years since the discovery that the Lewy pathology in PD and DLB neurons is immunoreactive for aSyn and, at the same time, neuronal aSyn accumulates in glial inclusions within MSA oligodendrocytes. Even though a plethora of studies focuses on the role of aSyn in neuronal physiology and pathology, increasing amount of data reinforces the contribution of non-cell autonomous neuron-glial interactions in the initiation and progression of a-synucleinopathies. Microglia and astrocytes form the brain’s defense system against neurotoxic insults, become activated and release pro-inflammatory factors. However, uncontrolled activation results in chronic microgliosis and astrogliosis that may be detrimental and lead to neurodegeneration. Even more, the deposition of aSyn in oligodendrocytes impairs their myelinating activity and reduces neuronal trophic support, events that eventually result in neuronal demise. Remarkably, neurons, microglia, astrocytes and oligodendrocytes are all able to take up and clear extracellular aSyn; however, glial cells appear to be the most potent scavengers. The endocytosis of various aSyn species might be conformation-sensitive, cell- and receptor-type specific, adding further complexity in disease management.

Undoubtedly, better understanding of the mechanisms mediating the interaction between neurons and glial cells in a-synucleinopathies may provide insights into neuronal dysfunction and death and may uncover novel disease modifying therapies.

## Figures and Tables

**Figure 1 ijms-22-04994-f001:**
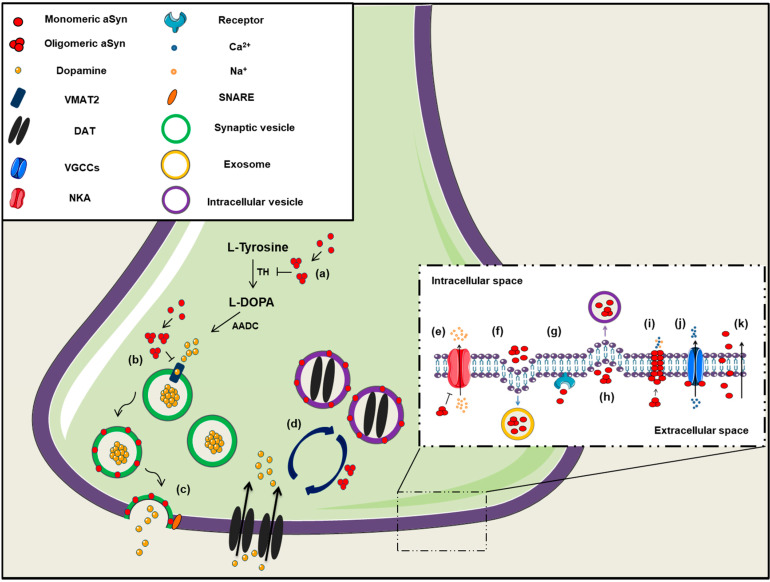
The role of aSyn at the presynaptic terminal. A schematic representation depicting of aSyn physiological and pathological effects at the synapse: (a) aSyn reduces the activity of tyrosine hydroxylase (TH), the enzyme responsible for catalyzing the conversion of L-Tyrosine to L-DOPA, thus impairing dopamine biosynthesis, (b) Increased levels of aSyn inhibit VMAT2, which is responsible for the uptake of monoamines (such as dopamine) into synaptic vesicles, (c) aSyn associates with synaptic vesicle membranes and regulates the SNARE-dependent vesicle fusion and neurotransmitter release, (d) Soluble aSyn interacts with the dopaminergic transporter DAT and decreases its amount on the plasma membrane, thus regulating the dopamine re-uptake from the synapse. However, aSyn aggregates trigger DAT recruitment to the plasma membrane that leads to massive entry of dopamine, (e) aSyn aggregates interact with Na^+^/K^+^-ATPase (NKA) preventing the effective pump out of Na^+^ ions, (f) aSyn is secreted from neuronal cells partly via associating with exosomes, (g) Extracellular aSyn interacts with neuronal receptors (i.e., LAG3) for its internalization in neurons or (h) it is up-taken via endocytosis, (i) PD-linked A30P and A53T mutant aSyn form large membrane pores through which most cations (i.e., Ca^2+^) can pass non-selectively, (j) Extracellular aSyn activates the voltage-gated Ca^2+^ channels (VGCCs), resulting in increased Ca^2+^ influx, (k) Monomeric aSyn enters neuronal cells via passive diffusion or direct penetration of their plasma membrane.

**Figure 2 ijms-22-04994-f002:**
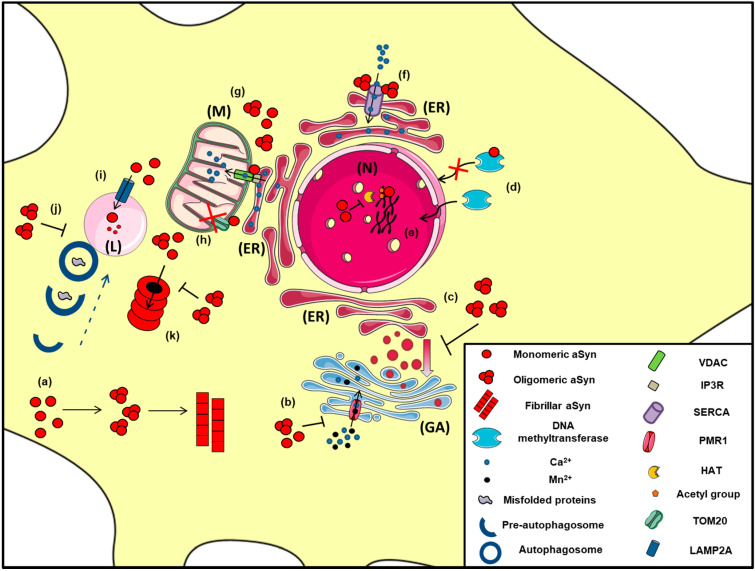
The proposed intracellular effects of various aSyn conformations in neurons. A schematic representation of the aberrant interactions between the various aSyn species with intracellular organelles: (a) In the cytoplasm of neurons, aSyn monomers form oligomers that can eventually become fibrils, (b) Both unfolded and aggregated aSyn impair the function of PMR1, a Ca^2+^-transporting ATPase pump that regulates Ca^2+^ and Mn^+2^ levels in the Golgi apparatus (GA), resulting in elevated cytosolic Ca^2+^ levels, (c) Both WT and mutant A53T aSyn disrupt the vesicular transport from Endoplasmic Reticulum (ER) to Golgi (GA), (d) WT aSyn inhibits the transportation of methyltransferases from the cytoplasm to the nucleus (N), thus altering DNA methylation of the *SNCA* gene, (e) Inside the nucleus (N), aSyn inhibits histone acetylation via its direct binding to histones or by preventing the action of histone acetyltransferase (HAT) enzymes, thus interfering in the process of gene transcription, (f) In the ER, aSyn aggregates activate the Ca^2+^-ATPase SERCA, resulting in dysregulated Ca^2+^ homeostasis, (g) Both monomeric and oligomeric aSyn interact with Voltage-dependent anion channel 1 (VDAC1) and inositol triphosphate receptors (IP3Rs), the protein components involved in mitochondrial-associated ER membrane (MAM) and regulates the transmission of Ca^2+^ signals from the ER to mitochondria (M), (h) aSyn binds to TOM20, a mitochondrial import receptor subunit and inhibits normal protein import, (i) Normally, monomeric or dimeric forms of aSyn are degraded in the lysosome (L) via Chaperone Mediated Autophagy (CMA), following their interaction with LAMP2A. However, under pathological conditions, impairment of CMA has been proposed to lead to aSyn accumulation and subsequent cell toxicity, (j) Oligomeric aSyn and various misfolded proteins are cleared via macroautophagy, following the fusion of autophagosomes with the lysosome. Pathological aSyn has been shown to inhibit autophagosome maturation or their fusion with lysosomes, thus impairing autophagic flux, (k) Monomeric and oligomeric aSyn are degraded via the proteasome; however, under pathological conditions, increased levels of aSyn or even soluble aSyn oligomers may inhibit proteasomal function, leading to aSyn accumulation and the formation of insoluble aggregates.

**Figure 3 ijms-22-04994-f003:**
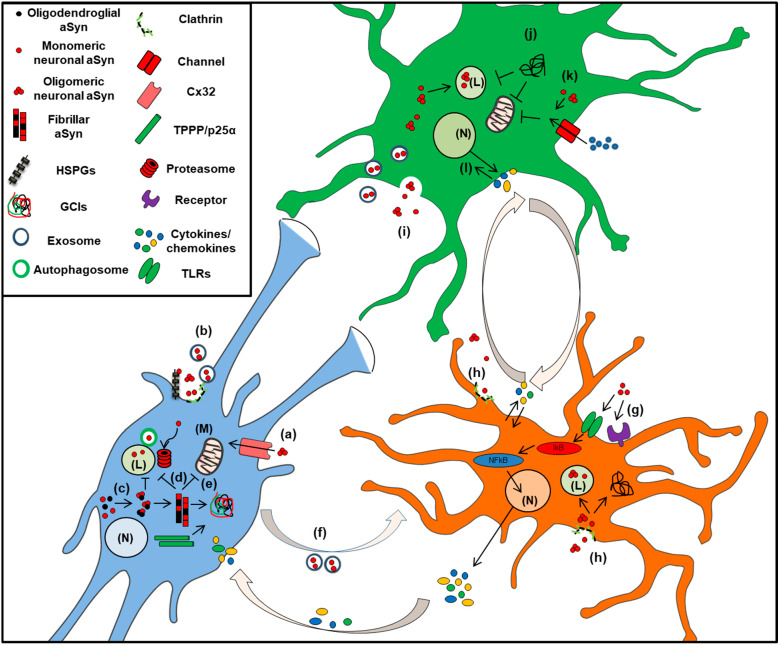
A complicated neuron-glial interrelationship. (a) Oligomeric aSyn enters oligodendrocytes (blue) via binding to the gap junction protein connexin-32 (Cx32), (b) Neuronally-derived aSyn enters oligodendrocytes via clathrin-mediated endocytosis, exosomal transportation, or via binding to Heparan Sulfate ProteoGlycans (HSPGs), (c) Inside oligodendrocytes, neuronal aSyn (red) initiates the seeding of the endogenous oligodendroglial aSyn (black) and together with the oligodendroglial-specific TPPP/p25α protein, they lead to the formation of GCIs, (d) aSyn aggregates impair the proteolytic machineries of oligodendrocytes [(proteasome and lysosome (L)], (e) Misfolded aSyn leads to mitochondrial (M) dysfunction and subsequent cell toxicity in MSA, (f) Oligodendroglial-derived exosomes containing aSyn have been shown to transfer to microglial cells (orange), (g) Extracellular aSyn stimulates TLRs and other receptors (i.e., FcγR, P2X7 etc) to activate microglial transcription factors, such as NFkB, for the production of various pro-inflammatory cytokines (IL1β, TNFa) and chemokines that induce astrocyte (green) reactivity and oligodendroglial damage, (h) Neuronally-secreted aSyn is taken up by microglial cells via clathrin-mediated endocytosis and is then driven to the lysosome (L) for degradation. In pathological conditions, though, it accumulates into aSyn insoluble aggregates, (i) Free or exosome-associated aSyn released by neurons is transmitted to astrocytes via endocytosis and enters the lysosome (L) for its clearance, (j) In disease, aSyn aggregates are formed that lead to lysosomal (L) impairment and mitochondria (M) dysfunction, (k) In astrocytes, aSyn triggers the opening of channels (i.e., Cx43 and Panx1), leading to dysregulation in Ca^2+^ homeostasis and altered mitochondrial morphology, (l) Microglia-secreted cytokines and chemokines activate astrocytes to further produce pro-inflammatory signaling molecules and enhance neurotoxicity.

**Figure 4 ijms-22-04994-f004:**
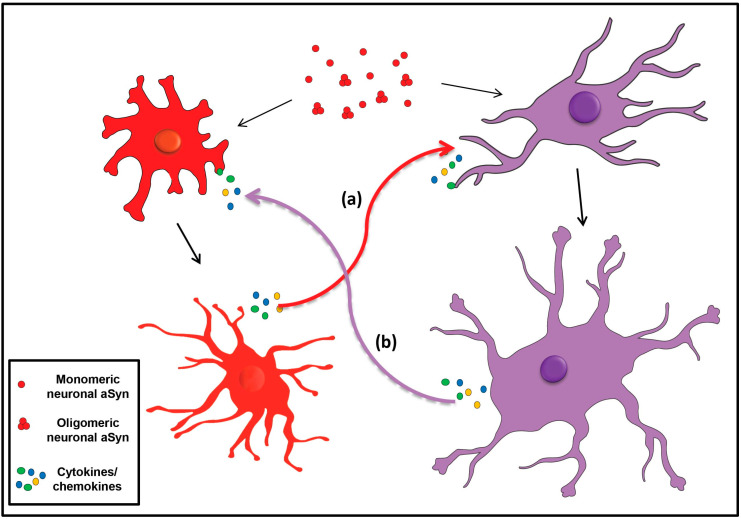
Crosstalk between astrocytes and microglia in a-synucleinopathies. Monomeric or oligomeric aSyn released by neurons is taken-up by astrocytes (purple) and microglia (red), which are then activated and secrete various cytokines or chemokines. (a) According to the prevailing hypothesis, the released pro-inflammatory molecules by microglia trigger the activation of astrocytes (red arrow), leading to excessive inflammation and neurotoxicity. (b) Conversely, aSyn can directly activate astrocytes to secrete pro-inflammatory cytokines or chemokines that recruit and activate microglia (purple arrow) resulting in excessive neuroinflammation.

## Data Availability

Not applicable.

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
