# Peer review of "Neurons and Glia Interplay in α-Synucleinopathies"

_ijms, 2021, doi:10.3390/ijms22094994_

Round 1

Reviewer 1 Report

In this Review 'Neurons and glia interplay in α-Synucleinopathies' the authors succinctly and comprehensively cover the current knowledge in this field. I have one general comment relating to the writing adhering to current dogma that microglia respond first and subsequently activate other glial cells. Numerous lines of investigation are currently challenging this view. To present a more balanced picture the authors should discuss the possibility including mechanisms and figures that questions the current dogma. For example a-Syn infected astrocytes have been shown to induce expression of Il-1 and Il-6 which may subsequently activate microglia. The current temporal order of these events is unknown but could occur reverse to the singular view presented in this review. Furthermore evidence for the Lidellow study quoted itself reveals astrocytes respond to LPS in Csf1r-/- mice (in the absence of microglia), similar to our own findings in prion disease infection https://www.biorxiv.org/content/10.1101/2021.01.05.425436v1. These data indicate that microglia polarize astrocytes but this process is separate from astrocyte activation - which in the case of α-Synucleinopathies (like prions) may derive from direct impacts of a-Syn on astrocytes independently of microglia. Acknowledgement of these data would make this review more cutting edge and relevant in terms of focus for example on targeting glial populations for therapeutic intervention. On a minor related point I found Figure 3 quite difficult to interpret based on the similar depiction of each cell type and their depicted location, I would suggest heavy revision of this figure to further clarify these points. 

Author Response

Reviewer 1

In this Review 'Neurons and glia interplay in α-Synucleinopathies' the authors succinctly and comprehensively cover the current knowledge in this field.

Response:We greatly appreciate the reviewer’s comments for our review.

  1. I have one general comment relating to the writing adhering to current dogma that microglia respond first and subsequently activate other glial cells. Numerous lines of investigation are currently challenging this view. To present a more balanced picture the authors should discuss the possibility including mechanisms and figures that questions the current dogma. For example a-Syn infected astrocytes have been shown to induce expression of Il-1 and Il-6 which may subsequently activate microglia. The current temporal order of these events is unknown but could occur reverse to the singular view presented in this review. Furthermore evidence for the Lidellow study quoted itself reveals astrocytes respond to LPS in Csf1r-/- mice (in the absence of microglia), similar to our own findings in prion disease infection https://www.biorxiv.org/content/10.1101/2021.01.05.425436v1. These data indicate that microglia polarize astrocytes but this process is separate from astrocyte activation - which in the case of α-Synucleinopathies (like prions) may derive from direct impacts of a-Syn on astrocytes independently of microglia. Acknowledgement of these data would make this review more cutting edge and relevant in terms of focus for example on targeting glial populations for therapeutic intervention.

Response: According to reviewer’s suggestion we have included a paragraph related to the activation of astrocytes prior and/or independent to microglia activation (see page 21, revised manuscript). We have also included a figure that describes the crosstalk between astrocytes and microglia in the presence of monomeric or oligomeric neuronally-derived aSyn (Figure 4 in the revised manuscript).

Regarding the reviewer’s comment on the Lidellow study (already cited in the original manuscript), our impression is that reactive microglia are required to induce A1 reactive astrocytes in vivo(as clearly mentioned in the Lidellow paper, page 3). This statement is based on their finding that astrocytes from Csf1r−/− mice failed to activate A1s and that purified astrocytes in culture, in the absence of Il-1α, TNFα, and C1q failed to respond to LPS (Extended Fig 2). So we think that this study does not challenge the current dogma and we have not included this study in this section.

  1. On a minor related point I found Figure 3 quite difficult to interpret based on the similar depiction of each cell type and their depicted location, I would suggest heavy revision of this figure to further clarify these points. 

Response:Following the suggestion of both reviewers we have modified Figure 3, omitting the effects of aSyn in neurons that are already mentioned in detail in Figures 1 & 2. We hope that the revised Figure 3 clarifies the main points of neuron-glia interactions.

Reviewer 2 Report

This review article aims to summarize the discoveries on alpha-synucleinopathies in neurons and glia. The authors outline all papers on the subject using in vitro and in vivo methods in the last 25 years. The conclusion is that the mechanisms mediating the interaction between neurons and glial cells in alpha-synucleinopathies are essential and need further exploration.

This manuscript falls on several counts:

  1. The object of a review is to critically appraise different studies to conclude something about old or new ideas that emerge from careful and thoughtful analyses of each paper examined. In this review, the authors simply state what the finding is in each paper and report it. They do not critically appraise, nor do they synthesize. Different models are probing the disease through different angles with different levels of approximation. Thus, conclusions from in vitro models should not be weighted equally as results from in vivo experiments. For instance, in the contradicting roles of aSyn nuclear localization, the listed researches supporting a protective role were primarily conducted in cell lines or yeast. In contrast, the majority of studies with both in vitro and in vivo models independently support a neurotoxic role. Furthermore, the kinds of genetic models (knockout, over-expression, or BAC transgenic) need to be considered.
  2. In several places, e.g.in proteasomal dysfunction in PD pathogenesis, the authors referenced the discoveries in post-mortem PD brains to support UPS's crucial role in the disease initiation/progression. While the importance of the discovery is not in dispute, the evidence needs further clarification. In that case, UPS function decrease with age, which complicated the post-mortem analysis. Also, in the late-stage neurodegeneration, multiple system dysfunctions concurred with aSyn inclusion.
  3. The diagram for neuron-glial interrelationship is exceptionally complicated. Although the authors depicted all the possible ways of transmitting aSyn between neurons and glial cells, they failed to emphasize the most important one (in the authors' opinion).
  4. Numerous PD-related aSyn mutants need some functional/structural introduction. Since the aSyn mutants such as A30P and A53T have been widely used and extensively referenced in the manuscript, the authors might want to discuss their structural/functional changes briefly.
  5. In line 981-982, inconsistent use of "," and "." in numbers.
  6. In line 172, citation required for "controversial results in the literature (19146388)"
  7. Repetitive wording: "on the other hand" 15+ times; "in agreement" 11+ times; "interestingly" 20+ times.

Author Response

Reviewer 2

This review article aims to summarize the discoveries on alpha-synucleinopathies in neurons and glia. The authors outline all papers on the subject using in vitro and in vivo methods in the last 25 years. The conclusion is that the mechanisms mediating the interaction between neurons and glial cells in alpha-synucleinopathies are essential and need further exploration.

This manuscript falls on several counts:

  1. 1. The object of a review is to critically appraise different studies to conclude something about old or new ideas that emerge from careful and thoughtful analyses of each paper examined. In this review, the authors simply state what the finding is in each paper and report it. They do not critically appraise, nor do they synthesize. Different models are probing the disease through different angles with different levels of approximation. Thus, conclusions from in vitro models should not be weighted equally as results from in vivo For instance, in the contradicting roles of aSyn nuclear localization, the listed researches supporting a protective role were primarily conducted in cell lines or yeast. In contrast, the majority of studies with both in vitroand in vivo models independently support a neurotoxic role. Furthermore, the kinds of genetic models (knockout, over-expression, or BAC transgenic) need to be considered.

Response: We thank the reviewer for his/her valuable comment. We have now followed the reviewer’s suggestion and commented on controversial findings in the literature. In regards to the comment “Thus, conclusions from in vitro models should not be weighted equally as results from in vivo experiments”, we believe that each model has a utility and provides unique information. Definitely the in vivomodels recapitulate better key concepts of a disease, but the in vitromodels can be proven more useful to provide mechanistic underpinnings. Nonetheless, we have taken into account the reviewer’s comment and modified our statements accordingly.

2.In several places, e.g.in proteasomal dysfunction in PD pathogenesis, the authors referenced the discoveries in post-mortem PD brains to support UPS's crucial role in the disease initiation/progression. While the importance of the discovery is not in dispute, the evidence needs further clarification. In that case, UPS function decrease with age, which complicated the post-mortem analysis. Also, in the late-stage neurodegeneration, multiple system dysfunctions concurred with aSyn inclusion.

Response: We agree with the reviewer’s comment that findings obtained from human post-mortem tissue should be treated with caution, especially in age-related diseases such as PD, given the overall decline in the function of multiple systems with aging. In the case of aSyn related pathology in a-Synucleinopathies, we believe that the use of tissue from affected and non-affected (in regards to aSyn pathology and neuronal death) brain areas may provide useful information regarding early or late events leading to neurodegeneration. We have included this comment in the revised manuscript.

3.The diagram for neuron-glial interrelationship is exceptionally complicated. Although the authors depicted all the possible ways of transmitting aSyn between neurons and glial cells, they failed to emphasize the most important one (in the authors' opinion).

Response: As mentioned above, we have modified Figure 3, omitting the effects of aSyn in neurons that are already mentioned in detail in Figures 1 & 2. We hope that the revised Figure 3 clarifies the main points of neuron-glia interactions.

In regards to the comment of emphasizing the most important way of aSyn transmission between neurons-glia cells, we have already mentioned in the original manuscript that the TLR-mediated endocytosis at least in microglia, is a key modulator for aSyn phagocytosis and subsequent propagation of alpha-synuclein pathology in glia cells (“Importantly, Toll-like-receptors 2 and 4 (TLR2 and TLR4) are considered crucial modulators of glial responses in a-synucleinopathies, as well as key players in aSyn phagocytosis”. page 17, 3d paragraph). In addition, we believe that transmission of aSyn through exosomes, represents also a key player in the neuron-glia propagation of aSyn pathology, as we already mention in the original manuscript (“Therefore, further considering the aforementioned involvement of exosomes in the transmission of aSyn pathology, targeting exosome-release from various cell types of the brain could be a potential therapeutic target against disease progression”, page 18, 4thparagraph).

4.Numerous PD-related aSyn mutants need some functional/structural introduction. Since the aSyn mutants such as A30P and A53T have been widely used and extensively referenced in the manuscript, the authors might want to discuss their structural/functional changes briefly.

Response: Following the reviewer’s suggestion we have included a paragraph introducing the most common PD-linked aSyn mutants in the revised manuscript (page 5, section 2.3. Aggregation and post-translational modifications).

5.In line 981-982, inconsistent use of "," and "." in numbers.

Response: This has been corrected.

6.In line 172, citation required for "controversial results in the literature (19146388)"

Response: This has been corrected and the citation is included.

7.Repetitive wording: "on the other hand" 15+ times; "in agreement" 11+ times; "interestingly" 20+ times.

Response: In line with the reviewer’s suggestion we have modified the writing of our manuscript to avoid repetitive wording.

Round 2

Reviewer 2 Report

The revised manuscript has been significantly improved from the original review.  The authors have provided more insights on the function and dysfunction of aSyn in the study.  There are meaningful updates on the diagram of aSyn interplay between oligodendrocytes, microglia, and astrocytes. Moreover, the addition of discussion on aSyn mutations makes the article more comprehensive.  It will be a valuable summary of the most up-to-date researches on alpha-synucleinopathies emphasizing the interplay between neurons and glia.